# Virtual Community: A Generative Social World for Embodied AI

## Abstract

We present Virtual Community, a social world simulation platform designed to support embodied AI research, featuring large-scale community scenarios derived from the real world. Virtual Community introduces two key features to enrich the virtual social world with generative AI: **scalable 3D Scene creation**, which supports the generation of expansive outdoor and indoor environments at any location and scale, addressing the lack of a large-scale, interactive, open-world scene for embodied AI research; and **embodied agents with grounded characters and social relationship networks**, the first to simulate socially connected agents at a community level, that also have scene-grounded characters. We design two novel challenges to showcase that Virtual Community provides testbeds to evaluate the social reasoning and planning capabilities of embodied agents in open-world scenarios: *Route Planning* and *Election Campaign*. The *Route Planning* task examines the agent's ability to reason about time, location, and tools in the community to plan fast and economical commutes in daily life. The *Election Campaign* task evaluates an agent's ability to explore and connect with other agents as a new member of the community. . We evaluate several baseline agents on these challenges and demonstrate the performance gap of current methods in addressing embodied social challenges within open-world scenarios, which our simulator is designed to unlock. We plan to open-source this simulation and hope Virtual Community can accelerate the development in this direction. We encourage the readers to view the demo of our simulation at https://sites.google.com/view/virtual-community-iclr.

## 1 Introduction

In recent years, we have witnessed tremendous progress in developing intelligent embodied agents, driven by advancements in embodied AI simulators (Savva et al., 2019; Puig et al., 2023b; Kolve et al., 2017; Li et al., 2021; Xiang et al., 2020b; Makoviychuk et al., 2021; Puig et al., 2018; Gan et al., 2021; Cheng et al., 2024). However, existing simulators face significant challenges in grounding realistic social interactions in 3D open-world environments. Most simulators focus on simulating a limited number of agents without incorporating social relationships (Szot et al., 2021; Gan et al., 2022), and are restricted to small-scale scene regions (Puig et al., 2018; 2020; 2023a;b; Zhang et al., 2023; 2024). In contrast, real-world social scenarios typically involve large communities of agents with diverse personalities and complex social networks spread over expansive areas. This limitation significantly restricts the study of complex and diverse social interactions between agents in simulated environments.

To address this challenge, it is crucial to enable the simulators to support the following key aspects. First, it should offer large-scale 3D environments, including complex and diverse indoor and outdoor scenes, capable of accommodating expansive agent communities and supporting tasks that span vast spatial regions. Current approaches for this aspect can be divided into manual design (Wang et al., 2024; Gan et al., 2021), which provide rich interactions but are inherently limited in number and diversity, and 3D reconstruction methods (Savva et al., 2019), which create visually realistic and diverse environments but often result in noisy scenes with limited interactivity in open-world settings.

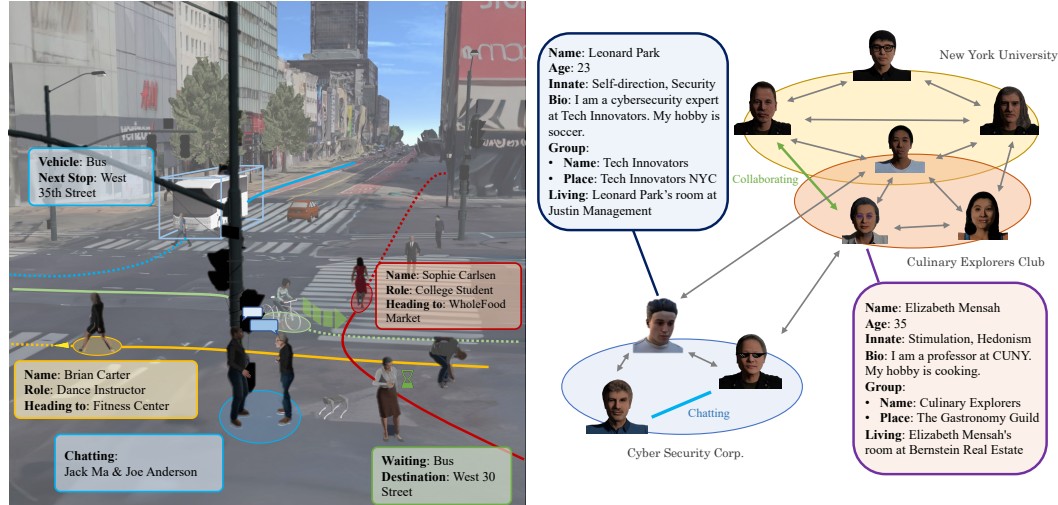

Figure 1: **Virtual Community features embodied agent communities within open-world scenes.** We provide a pipeline for automatically generating scenes and agent communities from real-world geospatial data. The agents are simulated in the Genesis physics engine as humanoid avatars, enabling them to engage in diverse social interactions within the community.

Second, the simulator must support a large number of interconnected 3D agents, each engaged in their own daily activities while maintaining social relationships with others, allowing for meaningful social interactions. However, existing multi-agent embodied AI simulators often lack the capacity to model complex social networks, limiting the study of rich social interactions. Additionally, they do not automatically align agent profiles, memories, and activities with the scene context, reducing the overall realism of the agent communities.

In this paper, we present Virtual Community, a generative social world for embodied intelligence research. Virtual Community addresses these challenges by integrating large-scale real-world geospatial data with generative models to produce interactive, scalable open-world scenes and socially grounded agent communities as shown in Figure 1. The platform is advanced in two aspects:

**Scalable 3D Virtual Scenes from Real-World Scenarios**     Virtual Community enables the fully automatic generation of 3D background scenes with several key features: (1) scalable indoor and outdoor scenes with customizable sizes and amounts, (2) automatic annotations of locations and objects within the scenes, and (3) a wide variety of interactive objects based on real-world locations. Virtual Community creates these scenes by combining generative models with real-world geospatial data, ensuring scalability in both data volume and scene size. Being built from geospatial data, these scenes can be seamlessly integrated with real-world tools including search engines and maps.

**Embodied Agents with Grounded Characters and Social Relationship Networks**     Virtual Community leverages open-world knowledge from foundation models to endow agents with rich, contextually grounded characters. The simulator further incorporates social relationship networks, connecting these agents into cohesive communities and enabling complex social interactions within the 3D environment. To support this, Virtual Community provides tens of human avatar skins integrated with SMPL-X skeletons, covering a diverse range of appearances, including celebrity likenesses, to ensure visual variety within human society. These avatars can perform over 15 distinct motions, such as walking, picking and placing objects, and operating vehicles, providing a broad spectrum of embodied behaviors.

Virtual Community uses Genesis[1], a generative physics simulator as the engine, which supports the simulation of a diverse range of materials and a vast range of robotic tasks while being fully

---

[1]https://github.com/Genesis-Embodied-AI/Genesis

differentiable. Genesis also comes with a real-time OpenGL-based renderer and a path tracing renderer implemented using Luisa compute (Zheng et al., 2022).

Virtual Community enables a variety of new possibilities in embodied AI research. The scalable scene generation and auto-annotation open a new challenge of open-world reasoning and planning, which we created a **Route Planning** challenge as a first step in this direction. The challenge involves the agent navigating from one geographic location to another, making decisions on transportation methods and on-road navigation. The generative embodied agent community provides the opportunity to study the social intelligence of embodied agents in complex and diverse social tasks in the open world. We propose the **Election Campaign**, which challenges agents to quickly familiarize themselves with other community members and persuade them to vote for the agent, testing their exploration and social communication skills.

Our simulator is novel in its ability to support long-duration and large-region tasks within real-world-based embodied AI simulators, marking a significant advancement in the field. By addressing the limitations of existing methods in data volume scaling, temporal scaling, and spatial scaling, we hope that our framework paves the way for training embodied general intelligence in environments that closely resemble the complexity and richness of the real world.

Table 1: Comparison of related simulation platforms

| Work | Real-world Setting | Social Networks | Multi-agent | Physics | Humanoid Action | Scalable Scene Size | Num Outdoor | Num Indoor |
|---|---|---|---|---|---|---|---|---|
| AI2-THOR | ✗ | ✗ | ✓ | ✓ | ✗ | ✗ | 0 | 120 |
| VirtualHome | ✗ | ✗ | ✓ | ✓ | ✓ | ✗ | 0 | 8 |
| Habitat 3 | ✗ | ✗ | ✓ | ✓ | ✓ | ✗ | 0 | 59 |
| iGibson | ✗ | ✗ | ✓ | ✓ | ✓ | ✗ | 0 | 15 |
| ThreeDWorld | ✗ | ✗ | ✓ | ✓ | ✓ | ✗ | 4 | ∞ |
| Minecraft | ✗ | ✗ | ✓ | ✗ | ✗ | ✗ | ∞ | ∞ |
| Carla | ✗ | ✗ | ✓ | ✓ | ✓ | ✗ | 12 | ✗ |
| Wayve | ✗ | ✗ | ✗ | ✗ | ✓ | ✗ | ∞ | ✗ |
| GPUtopia | ✗ | ✗ | ✓ | ✓ | ✓ | ✗ | 1 | 100K+ |
| **Virtual Community (Ours)** | ✓ | ✓ | ✓ | ✓ | ✓ | ✓ | ∞ | ∞ |

## 2 RELATED WORKS

### 2.1 EMBODIED AI SIMULATION

Recently, embodied AI has seen significant advancements through the development of simulation platforms. Most existing simulators primarily focus on household tasks within indoor environments (Beattie et al., 2016; Savva et al., 2019; Yi et al., 2018; Das et al., 2018; Xiang et al., 2020a; Shen et al., 2021; Szot et al., 2021; Li et al., 2021; Puig et al., 2018; Kolve et al., 2017; Yan et al., 2018), while some have extended support to outdoor scenes (Gan et al., 2021; Wang et al., 2024; Dosovitskiy et al., 2017; Kendall et al., 2018). However, these platforms lack diverse and scalable outdoor environments that can accommodate a larger number of agents and support more complex tasks. In contrast, this paper introduces a simulation platform featuring open-world environments with indoor and scalable outdoor scenes, enabling broader agent activities and more intricate task scenarios.

### 2.2 EMBODIED SOCIAL INTELLIGENCE

Current research on *Embodied Social Intelligence* is often limited to small agent populations in constrained household scenarios (Puig et al., 2020; Zhang et al., 2023; Stone et al., 2022) or simplified to 2D or grid worlds (Carroll et al., 2019; Suarez et al., 2019), hindering model development in the open world. Specifically, Park et al. (2023) demonstrates the robust simulation of human-like agents within a symbolic community, ignoring the 3D perception and realistic physics in the open world. Wang et al. (2023c) studies human-like simulation guided by system 1 processing with basic needs.

Predominant approaches, such as multi-agent reinforcement learning (MARL) and other planning models, face several limitations when applied to open-world settings. MARL, for instance, often struggles with scalability due to the exponential growth of state and action spaces as the number of agents increases (Wen et al., 2022). This makes it difficult to learn effective policies in complex, dynamic environments. Additionally, MARL approaches typically require extensive training data and computational resources, which may not be feasible in real-world applications. Other planning models, while potentially more efficient, often lack the adaptability required to handle the unpredictable nature of open-world interactions. They may rely on predefined rules or assumptions that do not hold in all scenarios, leading to suboptimal performance and limited generalization to new contexts (Puig et al., 2020).

### 2.3 FOUNDATION AND GENERATIVE MODELS FOR EMBODIED AI

With the recent advance of foundation models (Bubeck et al., 2023; Liu et al., 2023; Driess et al., 2023; Blattmann et al., 2023), numerous works have explored how they can help build powerful embodied agents (Wang et al., 2023b; Xi et al., 2023; Sumers et al., 2023; Wang et al., 2023d; Ahn et al., 2022; Sharma et al., 2021; Wang et al., 2023a; Park et al., 2023; Hong et al., 2024; Black et al., 2024), and scenes for simulation (Höllein et al., 2023; Schult et al., 2023; Deitke et al., 2022; Fu et al., 2021; Yang et al., 2024; Feng et al., 2024; Tang et al., 2023; Paschalidou et al., 2021). Robogen (Auerbach et al., 2014) leverages foundation models to automatically generate diversified tasks, scenes, and training supervision, thereby scaling up robotic skill learning with minimal human supervision. Different from them, this work aims to use a generative pipeline to create open world scenes and agent communities instead of constraint indoor scenes and tasks.

### 3 SCALABLE 3D SCENE GENERATION

The existing 3D geospatial datasets[2] provide extensive data in terms of quantity and diversity. However, they are not directly suitable for embodied AI research because of several limitations. First, these geospatial data often contain noise, including pedestrians, vehicles, and other transient objects that can disrupt simulations. Second, visual quality is inadequate for ground-level agent perspectives because these environments are typically reconstructed from aerial imagery, leading to less detailed textures and geometries at street level. To bridge this gap, we perform comprehensive mesh cleaning and enhancement in both geometry and texture to make the scenes suitable for embodied AI simulations.

To overcome these challenges, we propose a pipeline to transform 3D geospatial data into simulation-ready scenes for embodied AI. This pipeline consists of four main steps: mesh simplification, texture refinement, object placement, and automatic annotation. We list some qualitative example in Figure 4 and Figure 3

### 3.1 MESH CONSTRUCTION AND SIMPLIFICATION FOR SCENES

Since 3D geospatial data like Google 3D tiles are reconstructed from images using photometric methods, they often include noisy surfaces, excessive transient objects like moving cars and people, and unreliable mesh topology. These deficiencies make them inefficient and unsuitable for embodied AI simulations. To address this, we decompose the scene into the terrain, buildings, and decorative roofs and perform different operations to reconstruct each part of the scene.

The terrain is built procedurally using sparse reference elevation points and bilinear interpolation. We then derive simple and topologically sound mesh using information provided by the OpenStreetMap (OSM) service. The building mesh is then modified to better fit the Google 3D tiles geometry and to align with the terrain elevation. By aligning the mesh geometries with OSM primitives, we eliminate unnecessary details and artifacts, such as distorted surfaces and irregular shapes caused by aerial reconstruction errors. This geometric simplification not only reduces noise but also decreases the total number of primitives in the scene, leading to more efficient physical simulations and improved rendering performance.

---

[2]https://www.google.com/maps/

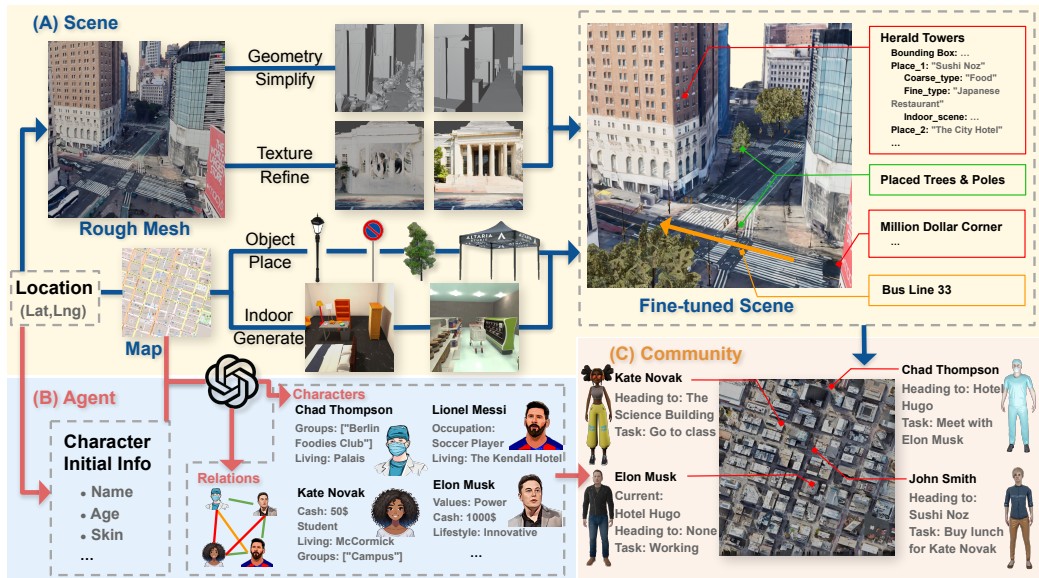

Figure 2: **Framework of the Virtual Community Generation Pipeline.** This pipeline generates scenes and corresponding agents from real-world geospatial data. The **scene generation** component (A) refines rough 3D data by using generative models to enhance textures and geospatial data to simplify geometry. It also utilizes generative methods to create interactive objects and detailed indoor scenes. The **agent generation** component (B) leverages LLMs to generate agent characters and social relationship networks based on scene descriptions, resulting in a socially grounded community of embodied agents (C).

## 3.2 ENHANCED TEXTURE QUALITY FOR REALISTIC SIMULATION

To improve the visual quality of 3D geospatial data, we employ advanced image processing techniques. First, we use an inpainting method based on Stable Diffusion (Rombach et al., 2022) to remove noise and repair missing or damaged areas in textures. This process corrects inconsistencies and eliminates artifacts from the reconstruction phase, resulting in smooth and realistic surface appearances. Once the texture integrity is restored, we enhance finer details using street view images and super-resolution tools. While some approaches directly use street view images for 3D reconstruction (Pang & Biljecki, 2022; Gao et al., 2024), these methods often struggle with limited coverage and density. Instead, we blend street view images with existing textures on scene primitives to improve visual richness. For super-resolution, we use GigaPixel [3] to increase texture resolution and sharpen finer details. This two-step enhancement significantly increases the visual fidelity of the textures, making them more suitable for ground-level rendering. These high-quality textures create a more immersive environment for agents, improving the realism and overall effectiveness of embodied AI training.

## 3.3 IMPROVING INTERACTIVITY BY OBJECTS RETRIEVAL AND REPLACEMENT

To enhance the interactivity of the scenes, we use generative methods to populate the environment with interactive objects, such as bikes and tent. We use annotations in OpenStreetMap (OSM) dataset to determine the type and location of generated objects to match the real-world context. The OSM annotations are used as input for the One-2-3-45 Liu et al. (2024) generative framework, which outputs the corresponding 3D meshes of corresponding objects. These generative objects are assigned physical properties that allow them to interact seamlessly with agents in the simulation. By aligning object generation with real-world geospatial data, this approach ensures that the scenes are functionally rich and physically interactive, enabling agents to engage in meaningful interactions that mirror real-world environments.

---

[3] https://www.topazlabs.com/gigapixel-ai

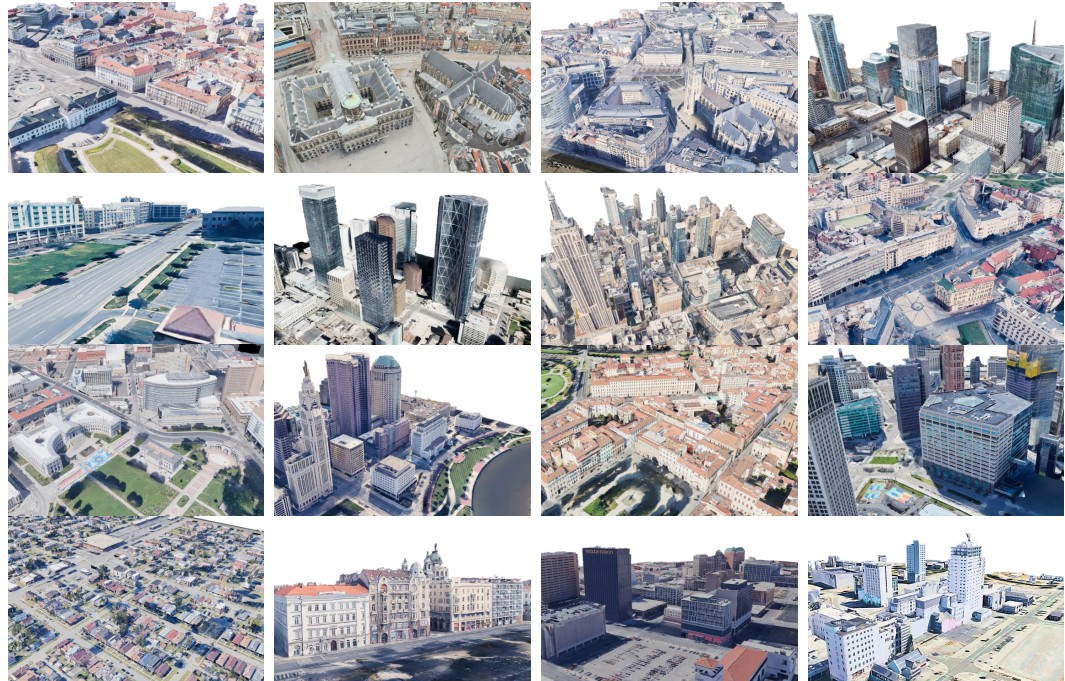

Figure 3: Large-scale scene rendered from the generated city scenes in North America and Europe. Our method is capable of generating high-quality scenes with an area of square miles. Objects are dynamically loaded in the simulator and are therefore not rendered in this figure.

### 3.4 AUTOMATIC ANNOTATION OF SCENES WITH GEOSPATIAL DATA

To facilitate alignment with real-world locations and provide semantic context, we automatically annotate the scenes using geospatial data. We integrate metadata from sources such as OpenStreetMap and other GIS databases to label buildings, roads, and other landmarks within the environment. This annotation enables agents to access location-specific information and supports tasks that require an understanding of the spatial context, such as navigation and location-based decision-making. The enriched semantic information enhances the potential for more sophisticated and context-aware agent behaviors within the simulation.

**Bus Transit Annotation** We search for the bus stops in the scene using Google Places API and then annotate the routes between any two adjacent bus stops using Google Directions API. We use depth-first search (DFS) on the graph of routes to find the route in the scene that maximizes the number of bus stops. Then, we decode polyline from the route to extract the dense waypoints which follow rounds of optimization to ensure the distance between waypoints is roughly equal and contains turning points. We also generate the bus schedule by estimating the travel time between bus stops using the distance and speed of the bus.

**Shared Bicycle Transit Annotation** We search for the shared bicycle stations in the scene using the OpenStreetMap API.

## 4 COMMUNITY OF EMBODIED AGENTS WITH GROUNDED CHARACTERS AND SOCIAL RELATIONSHIP NETWORKS

Given diverse generated scenes with real-world geospatial data mapping, we introduce a generative pipeline to populate the scenes with communities of agents with grounded characters and social relationship networks in section 4.1. Then we discuss how we design the embodiment for the agents and simulation details in section 4.2.

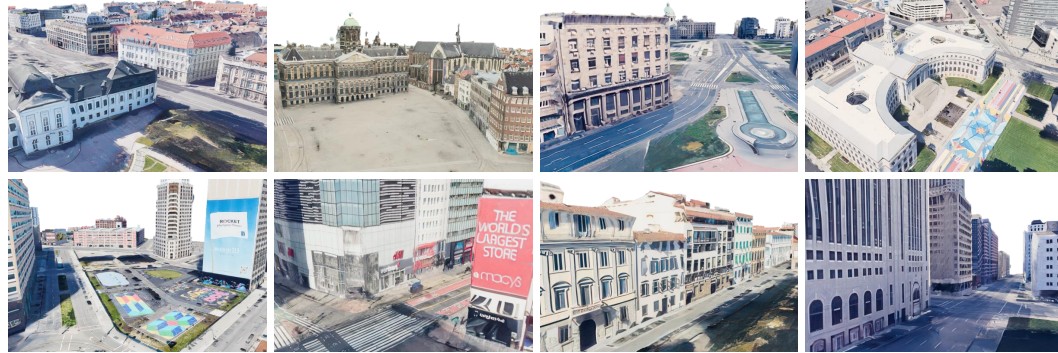

Figure 4: Close-up view of the generated scenes. The resulting scene has clean geometry and realistic texture, which is essential for physical simulation and sim-to-real transfer learning. Objects are dynamically loaded in the simulator and are therefore not rendered in this figure.

## 4.1 GROUNDED CHARACTERS AND SOCIAL RELATIONSHIP NETWORK GENERATION

We utilize the open-world knowledge of the Large Language Model (LLM) to generate agent character profiles and personalities grounded in the scene. The input to the LLM is structured into two parts to create characters grounded in a specific scene. The first part contains scene-related information, such as the scene name and details about various places, including their names, types, and functionalities. The second part includes details on the agents' appearances to ensure consistency between their visual attributes and generated profiles, which are annotated with the name and age. With both parts provided, the LLM generates agent profiles along with their social relationships. The profiles consist of basic attributes such as names, ages, occupations, personalities, and hobbies, which influence each agent's daily decision-making. Social relationships are structured as groups, each containing a subset of agents along with a text description and a designated place for group activities, connecting these agents into a cohesive community, and allowing rich and complex social interactions grounded in the 3D environment.

**Grounding Validator** We implemented a grounding validator to check if the generated character profiles are accurately grounded to the scene by checking if all related places generated exist in the scene. If the validation fails, LLM will be prompted again with the feedback from the validator and try to fix the mismatch. Empirically, we find that 1-2 rounds of prompting is enough to pass the grounding validator.

An example character with social relationship networks generated is shown in Figure 5 (a).

## 4.2 HUMAN AVATARS EMBODIMENTS

**Human Avatars Skin Creation** We obtained 12 avatar skin models of different genders, professions, and appearances from the Mixamo[4] website for integration into the Virtual Community. Each skin model of characters includes 71 skeletal joints and can be adapted to animation sequences in SMPL-X and FBX formats. To reduce the computational load during animation playback in the Virtual Community, we further optimized the skin models by applying Blender's Decimate Modifier tool, reducing the number of vertices in the 3D skin mesh by 90%.

In addition to the standard skin meshes provided by Mixamo, we use the Avatar SDK[5] to generate high-fidelity human skin meshes from real-world images, allowing us to represent diverse individuals, including celebrities, in our Virtual Community. For each character, we first obtain a high-quality portrait image from the internet. This image is processed using the Avatar SDK API, which produces a 3D mesh with detailed skin textures. To further enhance realism, we adjust the avatar's clothing, height, and body shape, creating a more lifelike and personalized appearance.

---

[4] https://www.mixamo.com/
[5] https://avatarsdk.com

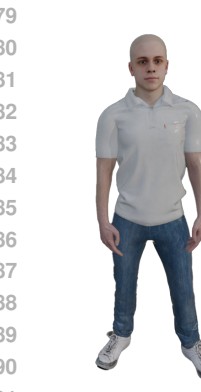

**Brian Carter**

Dance Instructor at Arthur Bodie Dance and Fitness Studio

**Age**: 35
**Values**: stimulation, hedonism
**Hobby**: dancing
**Group(s)**: CrossFit Warriors
**Living**: EŌS NoMad Apartments
**Cash**: $600
**Lifestyle**: I go to bed around 11pm, wake up around 7am, eat dinner around 8pm.

Daily Schedule

00:00:00 - 07:00:00
Sleep in Brian Carter's room
07:00:00 - 08:00:00
Morning Routine in Brian Carter's room
08:00:00 - 08:30:00
Commute
08:30:00 - 11:30:00
Dance Instruction in Arthur Bodie Dance and Fitness Studio
11:30:00 - 12:00:00
Commute
12:00:00 - 13:00:00
Lunch Break in Friedman's Herald Square
13:00:00 - 13:30:00
Commute
13:30:00 - 15:00:00
CrossFit Training in CrossFit NYC
......

Figure 5: An example of (a) generated character and (b) daily schedule.

**Human Avatars Motion Control** We combine SMPL-X human skeletons with created avatar skins to model human avatars in Virtual Community. The motions of these avatars are parameterized by SMPL-X pose vectors $J \in \mathbb{R}^{162}$ along with global translation and rotation vectors $T, R \in \mathbb{R}^3$. Based on these pose representations, a skin mesh for each avatar is calculated using forward kinematics.

Our motion model for humanoid avatars supports over 15 distinct motions, such as walking, picking objects, and entering various vehicles. We use motion clips from Mixamo and adjust these clips to our humanoid avatar models with appropriate animating speeds. For walking, we loop the walking motion clip until the avatar reaches the given distance. For object-related motions, the interacting object will be kinematically attached or detached to or from the avatars' hands depending on the action type. Similarly, the humanoid avatar will be kinematically attached or detached to or from the given vehicle for vehicle-related motions. We also incorporate physics constraints into our avatar motion model, where collision detection is performed between avatars and other scene entities, and the motion process is terminated when a potential collision is detected. To handle different terrain altitudes within community scenes, we preprocess a height field for each scene and kinematically adjust the height of our humanoid avatars according to their current locations.

### 4.3 DAILY SCHEDULE GENERATION

Given the scene-grounded characters and social relationship networks, we prompt the foundation models to generate the daily schedule for each agent, using a similar design to Park et al. (2023). Differently, we generate the daily schedule in a structured manner directly with each activity represented with a start time, an ending time, an activity description, and the corresponding activity place, and consider the required commute time between adjacent activities that are happening in different places explicitly, due to the actual cost of navigating in an expansive 3D environment. An example daily schedule generated is shown in Figure 5 (b).

## 5 OPEN WORLD AND SOCIAL CHALLENGES IN VIRTUAL COMMUNITY

We introduce and study two tasks in Virtual Community: *Route Planning* and *Election Campaigning*. The tasks cover agents planning ability in a community context and social intelligence to interact with other agents.

As the foundation for both tasks, agents in the community follow a default daily plan and routine (introduced in Section 4.3) if no specific tasks are assigned. During each episode, one or two agents are randomly selected and assigned one task. When an agent is given a task, it suspends its daily plan and focuses on completing the assigned social task in the community.

## 5.1 ROUTE PLANNING: USE TRANSPORTATIONS IN COMMUNITY

**Task Definition** To live a daily life in a human society, an embodied agent needs first to be able to plan its route from one place to another in the community. To study this basic ability of embodied agents, we introduce the *Route Planning* task. In this task, an agent needs to commute from place to place 5 - 7 times a day given the schedule. The agent can utilize available transit options, including buses with fixed routes and rental bikes along the roads. The bus is only available at the bus stop and the agent can only take a bus when the bus arrives. The bikes are available at given bike stations along the roads, and the agent also needs to return the bike to any bike station before the task finishes.

At each simulation step, agents are provided with an observation of RGB-D images with the corresponding camera matrix, current poses, daily schedules, and transit information in the community. The action space for these avatars includes *move_forward*, *turn_left*, *turn_right*, *enter/exit_bus*, and *enter/exit_bike*. The movement and turning actions can be set with a variable amount. When an agent is within a specified distance threshold of another agent, it can perform a communication action, enabling text-based interaction with agents within that range.

**Baselines** We compare three baseline agents in the *Route Planning* task:

•**Rule-based Agent** The rule-based agent always chooses to walk directly toward the target location.

•**MCTS agent** This agent is based on Monte Carlo Tree Search (MCTS) and simulates various decisions, such as choosing to take a bus. For each action, the agent estimates its associated cost and uses Monte Carlo sampling to iteratively update the expected reward for each decision path. The agent ultimately selects the action sequence that maximizes the cumulative reward.

•**LLM agent** This agent converts all the task information into a prompt and queries the Large Language Model (we use GPT-4o here) to generate a commute plan directly, which may include multiple steps such as walking to a bus stop, taking the bus to a specific stop, and then walk to the final destination.

All agents use the same low-level point-based navigation algorithm, which reconstructs the point cloud based on RGB-D images from the ego-centric observation at each step and converts the point cloud into a volume grid representation with a resolution of $0.1m$. Subsequently, a 2D occurrence map is established with a resolution of $0.5m$ based on this representation and an A* algorithm is used to search for the shortest path efficiently.

**Metrics** We evaluate the agents on two different scenes with 19 diverse personal schedules, making 106 commutes in total. Agents are expected to commute efficiently, so we use the following metrics for this task

• **Arrival Rate**: Percentage of in-time arrivals at the target location within the given time.

• **Time**: Average time in seconds taken on the road to reach the destination.

Table 2: Experiment results of *Route Planning* task.

| Methods | Route planning | |
| --- | --- | --- |
| | Arrival Rate↑ | Time↓ |
| Rule | 0.97 | 668.5 |
| MCTS | 0.91 | 698.7 |
| LLM (GPT-4o) | 0.89 | 963.0 |

**Results** As shown in Table 2, both search-based agent MCTS and LLM-based agent fail to make effective use of the available public transit options, resulting in even more time spent on commuting and a lower arrival rate compared to the naive rule agent baseline. This is due to the complexity of predicting whether the agent could catch a bus given partially built maps of the scene. We observe that the LLM agent tends to leverage public transits more but without a good estimation of the time needed to get to the transit station based on uncertainty on the navigation, it costs significantly more time in commuting.

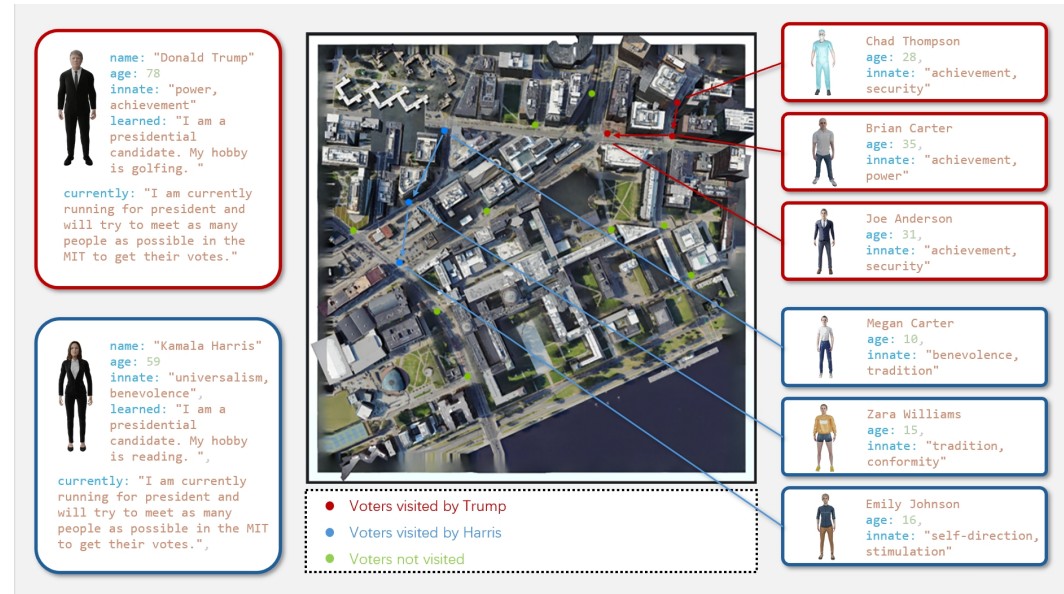

Figure 6: **Election Campaign Task Results.** We study LLM-driven agent behavior in the election campaign task. Different candidate agents exhibit distinct strategies.

## 5.2 ELECTION CAMPAIGN: FIND AND PERSUADE OTHERS

**Task Definition** In this task, two agents in the community are designated as candidates. The candidates need to navigate through the community, find potential voters, and persuade them to vote through direct communication. The election concludes at the end of the day, and the winner is determined by the percentage of votes each candidate receives. Due to pre-existing social relationships, some voters may have initial preferences for certain candidates at the beginning of the task, so candidates must devise strategies to influence and shift voter opinions throughout the election process.

**Baselines** We use an LLM-based agent as the baseline for this task. The agent's behavior is determined through iterative prompting of the LLM to identify which voter the candidate should visit next. After selecting the target voter, the candidate navigates to their location and delivers a campaign speech, also generated via LLM prompts. This process is repeated until the simulation ends. At the conclusion of the campaign, an election is held, during which each agent is prompted to decide which candidate they will vote for.

**Results** As shown in Figure 6, Trump began his campaign from the top right corner of the map, visiting Chad Thompson, Brian Carter, and Joe Anderson in succession. His target audience mainly consisted of achievement-oriented young men. In contrast, Harris started her actions from the top left corner of the map, visiting Megan Carter, Zara Williams, and Emily Johnson in succession. Her target audience is primarily focused on young women. Both chose targets that aligned with their campaign strategies.

## 6 CONCLUSION

We introduce Virtual Community, a generative social world for Embodied AI, featuring scalable scene generation and a community of embodied agents with grounded characters and social relationship networks. Virtual Community generation pipeline leverages rich real geospatial data and open-world knowledge data of advanced generative models and creates infinite scenes and grounded social agent communities. As an initial exploration of this simulator, we introduce two novel open-world and social challenges, **Route Planning** and **Election Campaign**, which are developed and tested using various baseline methods. These experiments highlight the difficulty of the challenges enabled by our new virtual social world. We hope Virtual Community can help advance the Embodied AI research towards building embodied generalist intelligence that can handle the difficulty of the real world and coexist with the human community.

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
