# OpenReview forum: "Virtual Community: A Generative Social World for Embodied AI"
_ICLR.cc/2025/Conference — ICLR 2025 Conference Withdrawn Submission_

### Official Review · Reviewer_Bftv · 2024-10-29

**Soundness:** 3
**Presentation:** 3
**Contribution:** 3
**Rating:** 5
**Confidence:** 4

**Summary:**

The paper presents Virtual Community, a generative social simulation platform designed for embodied AI research. It has two key contributions:
1. Scalable 3D Scene Creation: This feature enables the automatic generation of expansive indoor and outdoor environments using real-world geospatial data, filling a gap in open-world simulation for embodied AI.
2. Embodied Agents with Grounded Characters and Social Networks: The platform models agents with rich, grounded personalities and social networks, allowing for complex social interactions in large-scale, open-world settings.

The paper also introduces two novel tasks—Route Planning and Election Campaign—to assess the social reasoning and planning capabilities of embodied agents within the platform.

**Strengths:**

1. The paper is well-organized and uses visuals effectively to clarify the platform's components and features.
2. The platform models agents with grounded personalities and social networks, enabling the study of complex social interactions in a simulated community.
3. The automatic generation of detailed 3D environments from real-world geospatial data, combined with generative models, is really impressive.

**Weaknesses:**

1. While the paper claims support for large-scale environments, it lacks quantitative results on how the system scales in terms of computation, memory, or performance metrics (e.g., agent count or scene complexity). There is also no mention of whether the simulator offers a ‘headless mode’ to allow multiple instances to run simultaneously on a cluster, which would be valuable for generating large-scale data or evaluating models in parallel. Additionally, it is unclear whether the simulator can be run ‘faster than real-time’ to speed up experiments.

2. The paper doesn’t explain how the movements of characters not being controlled by the baseline algorithms are managed. It is unclear whether these movements are based on a social forces model, predefined behavior heuristics, or something else.

3. The generative approach to creating agent personalities and routines may not fully capture complex or realistic social behaviors.

4. The reliance on external geospatial data and APIs like Google Maps could limit applicability in areas with incomplete or unavailable data. Exploring methods for generating environments in a more data-independent manner would improve flexibility.

5. The paper doesn’t discuss the usability of the simulator or the available APIs for developers. It’s also unclear whether there’s a built-in way to model interactions as graphs, or if users would need to design these graphs themselves. Providing tools to model and track interactions as graphs would make it more user-friendly. Clear documentation and API examples would also help developers integrate the platform into their projects more easily.

6. There is no clear mention of the intended target audience for the simulator.

7. The paper omits references and comparisons to similar platforms like SEAN 2.0 [1] and Arena 1.0-4.0 [2], specifically along the axis of including the presence of robots in their simulator.

8. The platform focuses on simulation, but there is no discussion about how these simulated behaviors or learned policies can transfer to real-world applications.

[1] Tsoi, Nathan, et al. "Sean 2.0: Formalizing and generating social situations for robot navigation." IEEE Robotics and Automation Letters 7.4 (2022): 11047-11054.

[2] Shcherbyna, Volodymyr, et al. "Arena 4.0: A Comprehensive ROS2 Development and Benchmarking Platform for Human-centric Navigation Using Generative-Model-based Environment Generation." arXiv preprint arXiv:2409.12471 (2024).

**Questions:**

Refer to the Weaknesses section.

---

### Official Review · Reviewer_P3zP · 2024-11-01

**Soundness:** 2
**Presentation:** 3
**Contribution:** 2
**Rating:** 5
**Confidence:** 4

**Summary:**

The paper proposes Virtual Community, a 3D social open-world simulator to evaluate current embodied intelligence research. The authors build a pipeline to automatically generate 3D interactive city scenes, which transform 3D geospatial data into simulation-ready scenes, and enrich the simulation platform with embodied characters in SMPL-like format with character informations and social communities. To showcase the significance of this simulator, the authors propose two tasks: Route Planning and Election Campaign, to benchmark the involving progress of current embodied agent.

**Strengths:**

1. The automatic 3D interactive city scenes generation pipeline is quite interesting and insightful, which consists of mesh simplification, texture refinement, object placement, and automatic annotation.
2. It considers the rich personalities and social relationships that make up a community network, which may be meaningful for computational social science research.
3. The presentation and visualization of the paper is quite good.

**Weaknesses:**

1. While the authors claim that the pipeline can generate an infinite number of scenes as shown in Table 1, it is still important to note how many assets this work has generated so far. It lacks a quantitative analysis of these created assets.
2. The paper performs almost no quantitative experiments, only the proposed route planning task. This actually shows that the benchmark cannot cover too many current research methods, which is the biggest weakness of the paper.

**Questions:**

Since the paper uses human characters rather than actual humanoid or quadruped robots, what is the point of designing control spaces, physical properties, and interactive objects for them? It can be a navigation-only simulator.

---

### Official Review · Reviewer_E6Wc · 2024-11-02

**Soundness:** 2
**Presentation:** 2
**Contribution:** 1
**Rating:** 3
**Confidence:** 5

**Summary:**

This work addresses the use of AI (including LLM) to endow virtual agents with intelligent behavior, which includes locomotion, interaction and some form of transactions. The authors appear to emphasize the specific constraints of embodiment as well as the inclusion of social relationship between agents. The paper puts forward two integrated tasks to illustrate its technical proposal: one, path planning, of intermediate abstraction level and the other, election campaigning, with a more complex relation to agents personality and social interactions.

**Strengths:**

Incorporating AI techniques into virtual agents platforms and simulation has been a longstanding goal, faced with numerous fundamental as well as practical challenges. This paper is thus ambitious in its exploration of the problem and timely in the method chosen, which investigates the potential impact of LLM on this endeavor.

**Weaknesses:**

This type of work has seen a resurgence in interest since the development of LLM [Wan et al., 2024], although it has a long history featuring symbolic AI techniques of search-based path planning and plan-based decision making and behavior, of which not much has found its way to the paper’s reference list of analysis of previous work. Although more contemporary techniques certainly show promise including for this sort of application, this is unfortunate as i) LLM are generally weak on relevant topics such as planning (despite recents experiments such as [Nasir et al., 2024]) and ii) some VR integration problems have been previously described and it does not seem appropriate to re-invent them: in particular the relationship between AI-based action parameters and the lower-level control of action and animation in the virtual world.

Path planning is one of the two main tasks allocated to these virtual agents. However, the corresponding sections suffer from both a lack of clarity and a lack of technical details. In particular there is no clear articulation between the low-level A*-based path planning and higher level (MCTS, LLM). This is particularly confusing since there is a potential overlap between various levels, including at the level of AI techniques (see, e.g. [Xiao et al., 2023]). Moreover, since the authors only mention A*, they are not addressing the well-known issue of dynamic environments, which becomes increasingly relevant with the number of agents and autonomous vehicles.

Following Figure 2, there is a lack of details on how social relationships are established from the characters’ profiles, whose description is relatively simplistic. Reference to “open-world knowledge” does not appear sufficient in the light of the vats body of work dedicated to persona definition with LLM.

Results of Table II are difficult to interpret, in particular in terms of LLM performance. In addition, 19 personal schedules and 106 commutes can hardly be considered a large-scale simulation in view of previous work. Some explication given for the results, such as leveraging environmental resources, is simply not useful in the absence of a fair comparison of integration of high-level decision making and low level decision across the three techniques (Rules, MCTS, LLM).

Minor aside: There are some oddities in references, such as suggesting that Auerbach et al. [2014] would be a “foundational model”, in the sense of pretrained Transformers-based foundational models such as those listed in a previous section of the paper (Bubeck et al., 2023; Liu et al., 2023; Driess et al., 2023; Blattmann et al., 2023)


References

Wan, H., Zhang, J., Suria, A.A., Yao, B., Wang, D., Coady, Y. and Prpa, M., 2024, May. Building LLM-based AI Agents in Social Virtual Reality. In Extended Abstracts of the CHI Conference on Human Factors in Computing Systems (pp. 1-7).

Aghzal, Mohamed, Erion Plaku, and Ziyu Yao. "Can large language models be good path planners? a benchmark and investigation on spatial-temporal reasoning." arXiv preprint arXiv:2310.03249 (2023).

Song, C.H., Wu, J., Washington, C., Sadler, B.M., Chao, W.L. and Su, Y., 2023. Llm-planner: Few-shot grounded planning for embodied agents with large language models. In Proceedings of the IEEE/CVF International Conference on Computer Vision (pp. 2998-3009).

Xiao, H. and Wang, P., 2023. Llm a*: Human in the loop large language models enabled a* search for robotics. arXiv preprint arXiv:2312.01797.

Nasir, M.U., James, S. and Togelius, J., 2024. GameTraversalBenchmark: Evaluating Planning Abilities Of Large Language Models Through Traversing 2D Game Maps. arXiv preprint arXiv:2410.07765.

**Questions:**

How are social relations stablished from the agents' profiles. Are the actual agents' profiles more sophisticated than the examples given (e.g. through auto-generation or a memory for interactions, or evolving preferences?)
How are the various decision-making methods (Rules, MCTS, LLM) integrated with lower-level functions like path planning?
How are MCTS used in practice?

---

### Official Review · Reviewer_UQBA · 2024-11-04

**Soundness:** 3
**Presentation:** 2
**Contribution:** 3
**Rating:** 5
**Confidence:** 4

**Summary:**

This paper introduces a novel method for creating 3D scenes at scale. They show how to use this method to generate scenes in Route Planning and Election Campaign.

**Strengths:**

The idea of generating new simulation environments with Generative AI seems very useful as it would accelerate the pace at which we create and test out RL agents in more complex environments than possible today. Virtual Community is an big innovative step towards this.  The paper is quite clear in its presentation (pending some typos) that make it easy to understand the contributions and impact it could have on the community. If done well and deployed well, Virtual Community could be significant in driving innovations in the world of RL by forcing researchers to think about novel algorithms that are required to become experts in such environments.

**Weaknesses:**

Results section feels lacking. Only results for Route Planning are included; it would be nice to see results for the Election Campaign as well.
But more importantly, the election campaign environment doesn't seem very useful. Route planning requires having an environment and a simulation since the agent is actively taking actions in that environment. However, the election campaign environment is more about interactions with other people - not something that immediately requires having a 3D environment.

**Questions:**

1. Only MCTS and LLMs are included for baseline performance. A2C today is standard enough to where it should be an algorithm that is benchmarked to further show how having this environment would be helpful to the community.

2. How is the 3D environment helpful for the Election Campaign environment? Why is Virtual Community critical to generating this environment and what is an interaction that Virtual Community is enabling for this environment that wouldn't be possible with a simpler simulation of this?

3. If existing algorithms do so well on the current environments, what do you think are some unique insights that we could draw by using this environment and training RL agents on top of them? Is there an example of a "hard" environment that we could generate using Virtual Community.

4. I don't fully understand what the complete end to end process for building an environment with Virtual Community looks like. Would be great if this were better explained in the paper.


I am willing to change my rating if the above things are resolved!

---

### Note · Authors · 2024-11-14

I have read and agree with the venue's withdrawal policy on behalf of myself and my co-authors.